# COVID-19 Vaccination Effectiveness in the General Population of an Italian Province: Two Years of Follow-Up

**DOI:** 10.3390/vaccines11081325

**Published:** 2023-08-04

**Authors:** Annalisa Rosso, Maria Elena Flacco, Graziella Soldato, Giuseppe Di Martino, Cecilia Acuti Martellucci, Roberto Carota, Marco De Benedictis, Graziano Di Marco, Rossano Di Luzio, Matteo Fiore, Antonio Caponetti, Lamberto Manzoli

**Affiliations:** 1Department of Environmental and Prevention Sciences, University of Ferrara, 44121 Ferrara, Italy; annalisa.rosso@unife.it (A.R.); cecilia.martellucci@unife.it (C.A.M.); 2Local Health Unit of Pescara, 65124 Pescara, Italy; graziella.soldato@ausl.pe.it (G.S.); giuseppe.dimartino@ausl.pe.it (G.D.M.); roberto.carota@ausl.pe.it (R.C.); marco.debenedictis@ausl.pe.it (M.D.B.); graziano.dimarco@ausl.pe.it (G.D.M.); rossano.diluzio@ausl.pe.it (R.D.L.); antonio.caponetti@ausl.pe.it (A.C.); 3Department of Medical and Surgical Sciences, University of Bologna, 40100 Bologna, Italy; matteo.fiore7@studio.unibo.it

**Keywords:** SARS-CoV-2, COVID-19, vaccine, cohort study, Italy

## Abstract

We carried out a cohort study on the overall population of the province of Pescara, Italy, to assess the real-world effectiveness of SARS-CoV-2 vaccination against infection, severe, or lethal COVID-19, two years after the start of the vaccination campaign. We included all the resident or domiciled subjects, and extracted the official demographic, vaccination, COVID-19, hospital and co-pay exemption datasets from 1 January 2021, up to 15 February 2023. Cox proportional hazards analyses were adjusted for gender, age, diabetes, hypertension, COPD, major cardio- and cerebrovascular events, cancer, and kidney diseases. Throughout the follow-up (466 days on average), 186,676 subjects received greater than or equal to three vaccine doses (of ChAdOx1 nCoV-19, BNT162b2, mRNA-1273, NVX-CoV2373, or JNJ-78436735), 47,610 two doses, 11,452 one dose, and 44,989 none. Overall, 40.4% of subjects were infected with SARS-CoV-2. Of them, 2.74% had severe or lethal (1.30%) COVID-19. As compared to the unvaccinated, the individuals who received greater than or equal to one booster dose showed a ≥85% lower risk of severe or lethal COVID-19. A massive impact of vaccination was found among the elderly: 22.0% of the unvaccinated, infected individuals died, as opposed to less than 3% of those who received greater than or equal to three vaccine doses. No protection against infection was observed, although this finding was certainly influenced by the Italian restriction policies to control the pandemic. Importantly, during the Omicron predominance period, only the group who received at least a booster dose showed a reduced risk of COVID-19-related death.

## 1. Introduction

On 5 May 2023, the World Health Organization (WHO) recommended the United Nations declare an end to the COVID-19 crisis as a “public health emergency of International concern”, after a 12-month downward trend in the pandemic, characterized by a decrease in mortality and the easing of pressure on health systems [1].

Several vaccines were developed and administered to halt the pandemic, and the available evidence suggests that vaccination was able to confer a high protection against symptomatic COVID-19, with a good safety profile [2,3,4,5,6]. Although the primary two-dose schedule showed a decrease in vaccination’s protective effects over time, especially after the onset of the Omicron variant [7,8], several studies reported that vaccine booster doses were able to restore high levels of protection against COVID-19 severe outcomes [9,10,11,12,13,14,15,16,17,18,19,20,21] and, to a lesser extent, towards infection [11,22,23]. However, these studies are currently limited to a follow-up duration of less than six months [6,15,24], and an overall evaluation of the vaccine’s effectiveness on the general population after the start of the vaccination campaign is lacking.

We thus carried out a cohort analysis of the entire population of an Italian Province in order to compare the risk of SARS-CoV-2 infection and severe or lethal COVID-19 in vaccinated versus unvaccinated subjects that received one, two (primary schedule), or three/four doses (booster doses) during the first two years after the beginning of the vaccination campaign in January 2021.

## 2. Materials and Methods

This retrospective cohort study updates and expands two previous analyses, adding one year of follow-up [12,25]. We included all the permanent or temporary (domiciled) residents in the province of Pescara, Italy on 15 January 2021 (14 days after the beginning of the vaccination campaign) aged 10 years or more, without a positive SARS-CoV-2 swab at the date of follow-up start.

### 2.1. Data Sources

We extracted all the information from the following official National Healthcare System datasets, which are routinely collected, updated daily, and sent to the Italian Institute of Health [26]:-COVID-19 vaccinations (up to 31 December 2022);-SARS-CoV-2 PCR or rapid antigenic tests and COVID-19 cases (up to 15 February 2023, in order to allow a minimum of 45 days of follow-up);-Demographic (Italian “Anagrafica”, up to 15 February 2023).

To account for some of the main potential confounders of the association between vaccination and severe or lethal COVID-19 [27], we used (a) co-pay exemption database (Italian “Esenzioni Ticket”) and (b) the administrative discharge abstracts (Italian “SDO”), from the last ten years to extract the following conditions for each individual: diabetes (ICD-9-cm codes in any diagnosis field—250.xx); hypertension (401.xx–405.xx); major cardiovascular or cerebrovascular diseases—CVD (410.xx–412.xx; 414.xx–415.xx; 428.xx or 433.xx–436.xx); chronic obstructive pulmonary diseases—COPD (491.xx–493.xx); kidney diseases (580.xx–589.xx); and cancer (140.xx–172.xx or 174.xx–208.xx).

All the information from the above databases was merged through encrypted fiscal code.

### 2.2. Outcomes

We evaluated the effectiveness of SARS-CoV-2 vaccines to prevent the following:(a)infection with SARS-CoV-2, evaluated using RT-PCR; tested through nasopharyngeal swabs by the laboratories accredited by the region (throughout the follow-up), or using rapid antigen tests, performed by accredited laboratories or local pharmacies (only after January 2021);(b)COVID-19 severe disease, diagnosed by a specialist physician, virologically confirmed, and requiring hospital admission;(c)COVID-19-related death (inside or outside the hospital);(d)all-cause death.

### 2.3. Exposure—Vaccinations

According to the Italian National vaccination plan, BNT162b2, ChAdOx1 nCoV-19, mRNA-1273, JNJ-78436735, and (starting from March 2022) NVX-CoV2373 vaccines were gradually administered to the population, starting from 2 January 2021, free of charge [28,29].

From 2 January 2021 up to 31 December 2022, the subjects were divided into the following categories:(a)subjects who received no dose of vaccine were included in the reference group “unvaccinated”;(b)subjects who received only one dose of vaccine—including mRNA-1273, BNT162b2, ChAdOx1 nCoV-19, or NVX-CoV2373—were assigned to the group “1 dose”;(c)subjects who received only one dose of JNJ-78436735 vaccine, or only two doses of the other vaccines—BNT162b2, mRNA-1273, ChAdOx1 nCoV-19 or NVX-CoV2373—were assigned to the group “2 doses”;(d)subjects who received two or more doses of vaccines, if one of the administered vaccines was JNJ-78436735, or ≥3 doses of the other vaccines—BNT162b2, mRNA-1273, ChAdOx1 nCoV-19, or NVX-CoV2373—were assigned to the group “≥3 doses”.

For the analyses comparing the effectiveness of the diverse vaccine types, the individuals who received two or more different vaccines were included in a global category named “mixed vaccines”.

### 2.4. Follow-Up

The follow-up start varied by vaccination status:(a)16 January 2021, for the group “unvaccinated”;(b)two weeks after the single vaccination (to account for seroconversion time) for the group “1 dose”;(c)two weeks after the second dose of vaccine for the group “2 doses”;(d)two weeks after the third dose of vaccine for the group “≥3 doses”.

The follow-up ended the day in which the outcome was diagnosed (including death), or on 15 February 2023 for those with no outcomes.

### 2.5. Statistical Analyses

The risk of each outcome was separately compared between each vaccine group and the unvaccinated individuals. The analyses for the outcomes “COVID-19” and “COVID-19-related death” were restricted to the subjects who had at least one positive SARS-CoV-2 swab during the follow-up. Univariate analyses were stratified by age class (0–29 y, 30–59 y, 60+ y), gender, and type of vaccine. We choose these age categories to be consistent with the reports of the Italian Institute of Health [30] and with the Italian Government [31], which identified subjects aged ≥60 years as priority targets for immunization. The vaccination status (and, consequently, number of subjects in each vaccination group) may vary according to the outcome depending on the date in which the outcome occurred (before or after each vaccine dose). For example, if a subject who received three vaccine doses had a positive SARS-CoV-2 swab after the second dose and before the third dose, he/she was included in the group “two doses only” for the analyses of the outcome “infection”, while he/she was included in the group “three or four doses” for the outcome “all-cause death”. As the outcome “death” included all vaccine doses (as all were obviously administered before), the numbers in the descriptive Table (Table 1) are referred to the analyses of this outcome.

We used Cox proportional hazards analysis to compute the adjusted hazard ratios (HRs) of all outcomes by immunization status [32]. All the above-mentioned potential confounders (gender, age, hypertension, diabetes, COPD, CVD, kidney diseases, cancer, and infection (for all-cause death only)) were kept in all models a priori. To simulate a healthy population control, the main analyses were repeated restricting the sample to the subjects with none of the selected comorbidities and no hospitalization for any cause during the triennium preceding the pandemic (years 2018–2020—“healthy population”). The main analyses were also repeated restricting the follow-up to the Omicron predominance period (approximately from 1 January 2022 to the end of follow-up). In these analyses, the subjects who died or were infected with SARS-CoV-2 before 1 January 2022 were excluded, and the follow-up started on 1 January 2022 for all the subjects who were not vaccinated or had their last dose administered before that date. Moreover, multivariable analyses were also repeated for samples restricted to the subjects with hypertension, diabetes, or cancer. Finally, multivariable analyses stratified by vaccine doses were carried out to compare the effectiveness of the three most frequently administered vaccine schedules: BNT162b2, mRNA-1273, and mixed. Unvaccinated subjects were excluded from these analyses, which compared directly the likelihood of each outcome among those who received only mRNA-1273 vaccine or mixed vaccines, versus those who received only BNT162b2 (as the reference category). For all Cox models, the validity of constant incidence ratios over the follow-up was evaluated using Nelson–Aalen cumulative hazard estimates, and Schoenfeld’s test was used to assess the validity of the proportional hazards assumption [25]. Significance was set at <0.05 for all the analyses, which were carried out using Stata, version 13.1 (Stata Corporation, College Station, TX, USA, 2014).

## 3. Results

### 3.1. Characteristics of the Sample

The STROBE flowchart of the study participants is shown in Figure 1. From 2 January 2021 to 15 February 2023, a total of 290,727 residents or individuals domiciled in Pescara Province, Italy were included in the vaccine effectiveness analyses. As reported in Table 1, a total of 11,452 subjects received only one dose (3.9% of the population), 47,610 received two doses, completing the primary vaccination schedule (16.4%), and 186.676 received three or more doses of vaccine (64.2%), while 44,989 of the included subjects were not vaccinated against COVID-19 (15.5%). The mean age of the sample was 48.9 ± 20.8, and 48.8% were males. Overall, the prevalence of hypertension and diabetes mellitus were 13.9% and 5.4%, respectively. The majority of subjects who only completed the primary vaccination schedule received two doses of BNT162b2 (68.5%), while most individuals who received one or more booster doses received mixed vaccines (50.4%). The average follow-up was 466 ± 148 days.

### 3.2. SARS-CoV-2 Infection (Table 2)

During the follow-up, 40.4% of the study population had at least one positive SARS-CoV-2 swab (*n* = 117,559): 38.9% of the unvaccinated, 34.4% of those receiving one dose, 60.0% of those receiving two doses, and 35.7% of who received three or four doses. The incidence of SARS-CoV-2 infection varied by age and gender, with lower rates among the elderly (33.6% vs. 43.5% among the younger subjects) and the males (38.1% vs. 42.7% among the females).

**Table 2 vaccines-11-01325-t002:** SARS-CoV-2 infection incidence during the follow-up, overall and by vaccination status, gender, age, and vaccine type.

	Unvaccinated	1 Dose ^A^	2 Doses ^B^	3/4 Doses ^C^	*p*-Value ^G^	Total Sample
Overall sample ^D^, *n*	52,106	6341	50,359	181,921		290,727
Positive swabs, % (*n*)	38.9 (20,283)	34.4 (2181)	60.0 (30,221)	35.7 (64,874)		40.4 (117,559)
Gender						
Males, *n*	26,966	3229	24,830	86,932		141,957
Positive swabs, % (*n*)	35.7 (9627)	31.8 (1027)	56.7 (14,072)	33.7 (29,306)		38.1 (54,032)
Females, *n*	25,140	3112	25,529	94,989		148,770
Positive swabs, % (*n*)	42.4 (10,656)	37.1 (1154)	63.3 (16,149)	37.4 (35,568)	^V^	42.7 (63,527)
Age-class						
10–29 years, *n*	13,652	2160	17,249	30,113		63,174
Positive swabs, % (*n*)	43.7 (5964)	34.9 (754)	62.2 (10,722)	33.6 (10,106)	^V^	43.6 (27,546)
30–59 years, *n*	27,310	2978	24,485	82,102		136,875
Positive swabs, % (*n*)	38.3 (10,447)	38.1 (1133)	63.8 (15,617)	39.4 (32,382)	^I; V^	43.5 (59,578)
60+ years, *n*	11,144	1203	8625	69,706		90,678
Positive swabs, % (*n*)	34.8 (3872)	24.4 (294)	45.0 (3882)	32.1 (22,387)		33.6 (30,435)
Vaccine type ^E^						
BNT162b2, *n*	--	3522	33,591	70,110		107,223
Positive swabs, % (*n*)	--	35.1 (1237)	60.8 (20,406)	35.7 (24,991)	^V^	43.5 (46,634)
mRNA-1273, *n*	--	2423	10,583	20,150		33,156
Positive swabs, % (*n*)	--	30.1 (729)	49.6 (5248)	34.2 (6886)		38.8 (12,863)
Mixed vaccines ^F^, *n*	--	--	3789	91,661		95,450
Positive swabs, % (*n*)	--	--	76.6 (2904)	36.0 (32,997)		37.6 (35,901)

^A^ Please see Table 1 footnote A. ^B^ Please see Table 1 footnote B. ^C^ Please see Table 1 footnote C. ^D^ The number of subjects in each vaccine group varied according to the outcome, depending on the date in which the outcome occurred (before or after each vaccine dose). As an example, if a subject who received three vaccine doses had a positive SARS-CoV-2 swab after the second dose but before the third dose, he/she was included in the group “two doses only” for the analyses of the outcome “infection”, and in the group “three or four doses” for the outcome “death”. As the outcome “death” included all vaccine doses (as all were administered before), the numbers in the Table are referred to the analyses of this outcome. ^E^ The stratified analyses of the other vaccines (ChAdOx1 nCoV-19, JNJ-78436735, or NVX-CoV2373) have not been shown because of the scarcity of the sample. ^F^ Subjects who received at least two different vaccines. ^G^ Chi-squared test for the comparisons between the following: (I) unvaccinated vs. 1 dose group; (II) unvaccinated vs. 2 doses group; (III) unvaccinated vs. ≥3 doses group; (IV) 1 dose vs. 2 doses group; (V) 1 dose vs. ≥3 doses group; (VI) 2 doses vs. ≥3 doses group. When not reported, *p*-values were <0.05.

In the Cox proportional hazards analysis (Table 3), after controlling for gender, age, and selected comorbidities, the probability of a SARS-CoV-2 infection was significantly higher for all the categories of vaccinated subjects. Compared with the unvaccinated individuals, the subjects who received one, two, or three or more doses showed the following adjusted Hazard Ratios (HRs), respectively: 1.26 (95% CI 1.21–1.32), 2.41 (95% CI 2.37–2.46), and 1.27 (95% CI 1.25–1.29). The results did not substantially vary by age class or when the analyses were restricted to the “healthy” population (the subset of the population without any of the selected comorbidities and no hospitalization for any cause during the triennium preceding the pandemic—Appendix A), to the subset of participants with hypertension, diabetes, or cancer (Appendix A).

### 3.3. Severe COVID-19 (Table 4)

Overall, 2.74% of the population (*n* = 3226) developed severe COVID-19. The incidence of serious disease largely varied by vaccination status, being highest among the unvaccinated (7.96%), intermediate among those who received a single vaccine dose (3.07%), and lowest among the subjects who received two (1.59%) or more (1.64%) doses. The same trend emerged in all age classes and both genders. COVID-19 incidence differed slightly by gender but widely varied by age, being much higher (7.60%) among the elderly, as compared to the youngest (0.21%). Overall, the frequency of severe disease was lowest among those who received three or more doses of different vaccines (0.40%).

**Table 4 vaccines-11-01325-t004:** Severe COVID-19 during the follow-up, overall * and by vaccination status, gender, age, and vaccine type.

	Unvaccinated	1 Dose ^A^	2 Doses ^B^	3/4 Doses ^C^	*p*-Value ^G^	Total Sample
Overall sample * ^D^, *n*	20,283	2180	30,219	64,877		117,559
COVID-19, % (*n*)	7.96 (1615)	3.07 (67)	1.59 (481)	1.64 (1063)		2.74 (3226)
Gender						
Males, *n*						
COVID-19, % (*n*)	9627	1027	14,072	29,306		54,032
	9.40 (905)	2.83 (29)	1.71 (240)	1.84 (539)		3.17 (1713)
Females, *n*						
COVID-19, % (*n*)	10,656	1154	16,149	35,568		63,527
	6.66 (710)	3.29 (38)	1.29 (241)	1.47 (524)		2.38 (1513)
Age-class						
10–29 years, *n*	5964	754	10,722	10,106		27,546
COVID-19, % (*n*)	0.57 (34)	0.13 (1)	0.12 (13)	0.11 (11)	^IV; V^	0.21 (59)
30–59 years, *n*	10,447	1133	15,617	32,382		59,578
COVID-19, % (*n*)	5.34 (558)	1.50 (17)	0.77 (121)	0.49 (159)		1.44 (855)
60+ years, *n*	3872	294	3882	22,387		30,435
COVID-19, % (*n*)	26.4 (1023)	16.7 (49)	8.94 (347)	3.99 (893)		7.60 (2312)
Vaccine type ^E^						
BNT162b2, *n*	--	1237	20,406	24,991		46,634
COVID-19, % (*n*)	--	3.56 (44)	1.68 (342)	2.48 (621)		2.16 (1007)
mRNA-1273, *n*	--	729	5248	6886		12,863
COVID-19, % (*n*)	--	1.65 (12)	0.99 (52)	1.07 (74)		1.07 (138)
Mixed vaccines ^F^, *n*	--	--	3789	91,661		95,450
COVID-19, % (*n*)	--	--	1.32 (50)	0.40 (368)		0.44 (418)

* Analyses restricted to the infected subjects. ^A^ Please see Table 1 footnote A. ^B^ Please see Table 1 footnote B. ^C^ Please see Table 1 footnote C. ^D^ The number of subjects in each vaccine group varied according to the outcome, depending on the date in which the outcome occurred (before or after each vaccine dose). As an example, if a subject who received three vaccine doses had a positive SARS-CoV-2 swab after the second dose but before the third dose, he/she was included in the group “two doses only” for the analyses of the outcome “infection”, and in the group “three or four doses” for the outcome “death”. As the outcome “death” included all vaccine doses (as all were administered before), the numbers in the Table are referred to the analyses of this outcome. ^E^ The stratified analyses of the other vaccines (ChAdOx1 nCoV-19, JNJ-78436735, or NVX-CoV2373) have not been shown because of the scarcity of the sample. ^F^ Subjects who received at least two different vaccines. ^G^ Chi-squared test for the comparisons between: (I) unvaccinated vs. 1 dose group; (II) unvaccinated vs. 2 doses group; (III) unvaccinated vs. ≥3 doses group; (IV) 1 dose vs. 2 doses group; (V) 1 dose vs. ≥3 doses group; (VI) 2 doses vs. ≥3 doses group. When not reported, *p*-values were < 0.05.

The multivariable analyses confirmed the univariate results: in the total population, one, two, or three or more doses of vaccines were associated with a 53%, 73%, and 88% lower risk of severe COVID-19, respectively (all *p* < 0.001—Table 3). A similar dose–response trend was observed across all age classes, among the “healthy” subset of the population (Appendix A), and among the subjects with selected comorbidities (Appendix A).

### 3.4. COVID-19-Related Death (Table 5)

During the follow-up, 1523 of the infected subjects died, with an overall lethality rate of 1.30%. The proportion of COVID-19-related deaths did not substantially vary by gender, was lowest among those who received mixed vaccines (0.57%), and widely varied by age: 0.02% among the subjects aged 10–29 years (*n* = 6; all above 18 years of age), 0.20% among the adults aged 30-59 years (*n* = 120), and 4.59% among the elderly (*n* = 1397).

**Table 5 vaccines-11-01325-t005:** COVID-19-related deaths during the follow-up, overall * and by vaccination status, gender, age, and vaccine type.

	Unvaccinated	1 Dose ^A^	2 Doses ^B^	3/4 Doses ^C^	*p*-Value ^G^	Total Sample
Overall sample * ^D^, *n*	13,166	7292	27,472	69,629		117,559
Deaths, % (*n*)	4.38 (577)	0.88 (64)	0.84 (232)	0.93 (650)		1.30 (1523)
Gender						
Males, *n*	5998	3648	12,771	31,615		54,032
Deaths, % (*n*)	4.87 (292)	0.85 (31)	0.94 (120)	0.99 (314)		1.40 (757)
Females, *n*	7168	3644	14,701	38,014		63,527
Deaths, % (*n*)	3.98 (285)	0.91 (33)	0.76 (112)	0.88 (336)		1.21 (766)
Age-class						
10–29 years, *n*	3983	2286	10,023	11,254		27,546
Deaths, % (*n*)	0.10 (4)	0.00 (0)	0.00 (0)	0.02 (2)	^I; V; VI^	0.02 (6)
30–59 years, *n*	6854	3867	14,179	34,678		59,578
Deaths, % (*n*)	0.88 (60)	0.23 (9)	0.09 (13)	0.11 (38)		0.20 (120)
60+ years, *n*	2329	1139	3270	23,697		30,435
Deaths, % (*n*)	22.0 (513)	4.83 (55)	6.70 (219)	2.57 (610)		4.59 (1397)
Vaccine type ^E^						
BNT162b2, *n*	--	3679	19,426	27,026		50,131
Deaths, % (*n*)	--	1.20 (44)	0.90 (175)	1.41 (380)		1.19 (599)
mRNA-1273, *n*	--	3532	5163	7271		15,966
Deaths, % (*n*)	--	0.54 (19)	0.81 (42)	0.92 (67)	^IV; V^	0.80 (128)
Mixed vaccines ^F^, *n*	--	--	1219	35,332		36,551
Deaths, % (*n*)	--	--	0.49 (6)	0.57 (203)		0.57 (209)

* Analyses restricted to the infected subjects. ^A^ Please see Table 1 footnote A. ^B^ Please see Table 1 footnote B. ^C^ Please see Table 1 footnote C. ^D^ The number of subjects in each vaccine group varied according to the outcome, depending on the date in which the outcome occurred (before or after each vaccine dose). As an example, if a subject who received three vaccine doses had a positive SARS-CoV-2 swab after the second dose but before the third dose, he/she was included in the group “two doses only” for the analyses of the outcome “infection”, and in the group “three or four doses” for the outcome “death”. As the outcome “death” included all vaccine doses (as all were administered before), the numbers in the Table are referred to the analyses of this outcome. ^E^ The stratified analyses of the other vaccines (ChAdOx1 nCoV-19, JNJ-78436735, or NVX-CoV2373) have not been shown because of the scarcity of the sample. ^F^ Subjects who received at least two different vaccines. ^G^ Chi-squared test for the comparisons between: (I) unvaccinated vs. 1 dose group; (II) unvaccinated vs. 2 doses group; (III) unvaccinated vs. ≥3 doses group; (IV) 1 dose vs. 2 doses group; (V) 1 dose vs. ≥3 doses group; (VI) 2 doses vs. ≥3 doses group. When not reported, *p*-values were <0.05.

SARS-CoV-2 lethality was much higher among the unvaccinated (4.38%) than all vaccinated subjects, whose lethality rate slightly varied from 0.84% to 0.93% depending on the number of doses. The results were similar across genders. The difference in lethality between vaccinated and unvaccinated individuals was massive among the subjects aged 60 or more years: 22.0% of the unvaccinated, infected individuals died, as opposed to less than 3% of those who received three or more vaccine doses. Almost half (48.6%) of the 617 infected, unvaccinated subjects aged 80 years or more died during the follow-up (10.4% died among the vaccinated).

Multivariable analyses again confirmed the univariate results: among the infected individuals, the subjects who received one, two, or three or more vaccine doses showed a 64%, 62%, and 85% reduction in the risk of death, respectively, as compared to the unvaccinated subjects (all *p* < 0.001—Table 3). The results were similar in the “healthy” subset of the population (Appendix A), among the subjects with hypertension, diabetes, or cancer (Appendix A), and were consistent for all individuals aged 30 years or more, while the analyses in the younger age class were underpowered due to the low number of events.

### 3.5. All-Cause Death (Table 6)

A total of 6821 individuals were deceased for any cause (2.38% of the population) during the follow-up. The death rate was almost identical for both genders, lower among those who received mixed vaccines (0.91%), and largely varied by age: 0.04% among the subjects aged 10–29 years, 0.34% among the adults aged 30–59 years, and 7.0% among the elderly. The overall mortality was highest among the unvaccinated individuals (4.41%), followed by those who received two vaccine doses (4.13%), those who received a single dose (2.84%), and the subjects who received three or more doses (1.36%).

**Table 6 vaccines-11-01325-t006:** All-cause deaths, overall and by vaccination status, gender, age, and vaccine type.

	Unvaccinated	1 Dose ^A^	2 Doses ^B^	3/4 Doses ^C^	*p*-Value ^G^	Total Sample
Overall sample ^D^, *n*	44,989	11,452	47,610	186,676		290,727
Deaths, % (*n*)	4.41 (1986)	2.84 (325)	4.13 (1964)	1.36 (2546)		2.35 (6821)
Gender						
Males, *n*	23,337	5850	23,529	89,241		141,957
Deaths, % (*n*)	3.93 (918)	2.85 (167)	4.18 (984)	1.44 (1287)	^II^	2.36 (3356)
Females, *n*	21,652	5602	24,081	97,435		148,770
Deaths, % (*n*)	4.93 (1068)	2.82 (158)	4.07 (980)	1.29 (1259)		2.33 (3465)
Age-class						
10–29 years, *n*	11,671	3692	16,550	31,261		63,174
Deaths, % (*n*)	0.11 (13)	0.11 (4)	0.01 (2)	0.02 (6)	^I; VI^	0.04 (25)
30–59 years, *n*	23,717	5712	23,047	94,399		136,875
Deaths, % (*n*)	0.74 (175)	0.65 (37)	0.46 (107)	0.18 (153)	^I; IV^	0.34 (472)
60+ years, *n*	9601	2048	8013	71,016		90,678
Deaths, % (*n*)	18.7 (1798)	13.9 (284)	23.2 (1855)	3.4 (2387)		7.0 (6324)
Vaccine type ^E^						
BNT162b2, *n*	--	5964	32,611	72,145		110,720
Deaths, % (*n*)	--	3.44 (205)	4.28 (1397)	1.92 (1383)		2.70 (2985)
mRNA-1273, *n*	--	5226	10,498	20,535		36,259
Deaths, % (*n*)	--	1.84 (96)	4.43 (465)	1.50 (308)	^V^	2.40 (869)
Mixed vaccines ^F^, *n*	--	--	2104	93,996		96,100
Deaths, % (*n*)	--	--	0.81 (17)	0.91 (855)	^VI^	0.91 (872)

^A^ Please see Table 1 footnote A. ^B^ Please see Table 1 footnote B. ^C^ Please see Table 1 footnote C. ^D^ The number of subjects in each vaccine group varied according to the outcome, depending on the date in which the outcome occurred (before or after each vaccine dose). As an example, if a subject who received three vaccine doses had a positive SARS-CoV-2 swab after the second dose but before the third dose, he/she was included in the group “two doses only” for the analyses of the outcome “infection”, and in the group “three or four doses” for the outcome “death”. As the outcome “death” included all vaccine doses (as all were administered before), the numbers in the Table are referred to the analyses of this outcome. ^E^ The stratified analyses of the other vaccines (ChAdOx1 nCoV-19, JNJ-78436735, or NVX-CoV2373) have not been shown because of the scarcity of the sample. ^F^ Subjects who received at least two different vaccines. ^G^ Chi-squared test for the comparisons between: (I) unvaccinated vs. 1 dose group; (II) unvaccinated vs. 2 doses group; (III) unvaccinated vs. ≥3 doses group; (IV) 1 dose vs. 2 doses group; (V) 1 dose vs. ≥3 doses group; (VI) 2 doses vs. ≥3 doses group. When not reported, *p*-values were <0.05.

At multivariable analyses (Table 3), a dual association between vaccination and all-cause death was observed: while those who received only one or two vaccine doses showed a significantly higher risk of all-cause death (HRs 1.40 and 1.36, respectively; both *p* < 0.001), the subjects who were administered three or more vaccine doses showed a substantially lower risk of death (HR 0.22; 95% CI: 0.20–0.23). Such a dual trend was observed only among the elderly, as no significant increase in the risk of all-cause death was detected among any of the vaccinated adult or younger populations. Similar results were observed in all subsets of the population (Appendix A).

### 3.6. Additional Analyses: Omicron Predominance and Vaccine Types

When the analyses were repeated restricting the follow-up to the Omicron predominance period (Appendix A), both univariate and multivariable results were concordant to those of the overall sample, with the notable exception of lower protection among the subjects who received a single or two vaccine doses only. Against COVID-19, the adjusted HRs of the recipients of one or two doses were 0.76 and 0.63, respectively (versus 0.47 and 0.27 during the entire follow-up). Most importantly, no protection was observed for these subjects against COVID-19-related deaths (both *p* > 0.05). Thus, during the Omicron predominance period, only the individuals who received at least one booster dose showed a significantly and substantially lower risk of lethal COVID-19.

The results of the direct comparison of the most frequently administered vaccines, stratified by dose, have been reported in Appendix A. After a single dose, as compared to the subjects who received the BNT162b2 vaccine, the individuals that received the mRNA-1273 vaccine showed a significantly lower likelihood of all outcomes except COVID-19 (*p* > 0.05). After two doses only, the differences across vaccines were mostly non-significant, with a few exceptions: as compared with BNT162b2 recipients (reference category), those who received mRNA-1273 or mixed vaccines showed, respectively, a lower and higher risk of infection, while all-cause deaths were less frequent in the mixed vaccines group. Finally, after three or more doses, all outcomes except infection were less frequent among mixed vaccine recipients, while the risk of both all-cause and COVID-19-related death was significantly higher among the subjects who received only mRNA-1273.

## 4. Discussion

This retrospective cohort study evaluated COVID-19 vaccination effectiveness in the general population of an Italian province two years after the start of the immunization campaign. The main findings are the following: (a) compared with the unvaccinated, adjusting for age, gender, selected comorbidities, and follow-up duration, the subjects who received any dose of COVID-19 vaccines showed a substantially lower risk of both severe COVID-19 disease and COVID-19-related death; (b) the greatest risk reduction (<80%) of COVID-19 and related death was observed among the individuals who received one or more booster doses, and also showed a substantially lower risk of all-cause death; (c) a massive impact of vaccination was found among the elderly, with an absolute risk reduction of COVID-19-related death as large as 19% between vaccinated and unvaccinated individuals; (d) vaccination was associated with a higher likelihood of SARS-CoV-2 infection; (e) during the Omicron predominance period, only the subjects who received at least one booster dose showed a significant (and substantial) reduction in the risk of COVID-19-related death; (f) no considerable differences in vaccination effectiveness were observed across genders.

Our results confirm and expand the current evidence on the effectiveness of COVID-19 vaccines in preventing the serious outcomes of SARS-CoV-2 infections [12,25]: those who received only one or two vaccine doses still showed a substantially lower risk than the unvaccinated, even after 12 months of follow-up. Moreover, booster doses were able to further halve the risk when compared to the standard vaccination schedule (adjusted HRs of booster versus two doses: 0.43 for both outcomes; *p* < 0.001), and a clear dose–response was observed. The impact of booster doses on the protection against severe disease or death was even larger when the analyses were restricted to the Omicron predominance period, when the effectiveness of the single or two doses schedules were substantially lower than during the overall follow-up, and only the subjects who received at least one booster dose showed a significant decrease in the risk of a lethal COVID-19. The latter findings on booster doses’ utility to restore protection from serious outcomes, especially against the Omicron variant, are certainly not novel [10,11,12,13,14,15,17,33], but none of the studies published so far had a follow-up longer than 6 months [6,10,11,15,24], and these findings provide significant data, although inevitably preliminary, about the duration of effect of booster doses. 

Given their higher risk of severe COVID-19-related outcomes [34,35,36], the finding that the highest absolute benefit of vaccination was observed among the elderly was clearly expected [37,38]. The magnitude of such benefit, however, after two years of follow-up, was impressive: once infected, more than one-fifth of the unvaccinated older subjects died, as compared to one out of twenty of those who received a booster dose. The absolute risk differences were 19% for COVID-19-related deaths and 22% for severe COVID-19 cases. In this age class, due to the combination of a highly effective vaccine (>80% in our study as well as in other studies [37,38]) and a disease with such a high case-fatality rate, the impact of vaccination was massive, and the prolonged follow-up provides a strong confirmation of the correctness of the public health entities which prioritized immunization among the interventions to control the pandemics [39]. As regards the younger subjects (<30 years of age), in line with available evidence on the lower severity of SARS-CoV-2 infection among this subset of the population, we observed a low absolute frequency of severe COVID-19 and related deaths in this age class (2 out 1000 and 2 out 10,000, respectively) [40]. However, even in this younger group, thanks to a high vaccine effectiveness, the overall impact of vaccination was still significant, as the absolute risk reduction of severe COVID-19 among the infected was 0.5%, confirming the protective role of vaccination against severe outcomes of infection found in other studies [41,42,43]. Finally, it is worth noting that the overall results on vaccine effectiveness, that were observed in the total sample, were similar to the results obtained when the analyses were restricted to the “healthy” population, or to the subjects affected by hypertension, diabetes, or cancer, in agreement with a previous population-based study from Spain [44]. 

The observed positive association between vaccination and SARS-CoV-2 infection needs to be interpreted with caution, since it might have been caused by very different behaviors of unvaccinated and vaccinated individuals, as a consequence of the restrictive measures adopted in Italy to control the epidemic. As previously discussed [12,23,25,45], a vaccination or recovery certificate (Green Pass), issued to people who received at least three doses or had a recent SARS-CoV-2 infection, was mandatory for travel, to retain a job, or access most public and private venues from July 2021 to May 2022 [46]. These measures inevitably lead to a major bias in the evaluation of this outcome: given the much higher mobility granted to vaccinated individuals, these individuals were much more likely to be exposed to contagion than the unvaccinated [47]. Moreover, as of February 2022, a SARS-CoV-2 test was no longer mandatory for asymptomatic close contacts of a confirmed case [48], and unvaccinated individuals may have been more likely to avoid testing than vaccinated subjects for ideological reasons, a different health-seeking behavior, or to avoid self-isolation [47,49,50]. In any case, even if it is highly unlikely to assume that vaccination increased the risk of infection, it certainly showed a suboptimal protection against contagion, as indeed reported by several previous studies [10,11,24].

A seemingly nonsensical finding was the higher risk of all-cause death observed among the subjects that received one or two vaccine doses, as opposed to the much lower mortality of those who received three or more doses. However, this has been described before [32] and can be explained again by the above-mentioned Italian policies to control the pandemics. In fact, due to the strong pressure to receive at least three doses, there were only two reasonable choices: (a) receive zero doses (for those who could not be vaccinated or refused the vaccine in toto) and (b) receive three doses (for all others, because one or two doses only were not sufficient to obtain the Green Pass). It is thus clear that many of the subjects who received only one or two doses were likely discouraged from further immunizations because of the occurrence of a disease or were deceased before the third dose, and sicker subjects were therefore selectively retained into the groups “1 dose” or “2 doses”, determining the large difference observed in all-cause death rates according to the vaccination dosage [32].

The comparison of different vaccines’ effectiveness provided mixed results, making it complex to draw clear conclusions. While several studies reported a higher effectiveness of mRNA-1273 compared to BNT162b2 against infection and severe COVID-19 outcomes both as a primary and booster vaccination [51,52,53,54,55,56], in the present cohort we observed a significantly higher protection of mRNA-1273 only against SARS-CoV-2 infections or after a single dose, and the subjects who received three or more doses of mRNA-1273 showed a higher likelihood of death, as compared to the recipients of BNT162b2 only. In contrast, we observed a greater protection conferred by heterologous booster (mixed) vaccination compared with a homologous booster with BNT162b2 only, in line with the findings of immunogenicity [57] and other population-based [20,58,59] studies. Notably, however, the results are difficult to compare due to the diverse booster schedules adopted across countries.

As regards gender differences, higher rates of SARS-CoV-2 infection were found among females, while a slightly higher risk of severe COVID-19 was observed among males, in line with available evidence on a higher susceptibly of men to worse outcomes of infection [27,60,61]. However, no significant differences in vaccine effectiveness were observed for all outcomes across genders, as previously reported by other studies [62,63], with a similar dose–response trend for both genders. The need for gender-disaggregated data on the effectiveness and safety of COVID-19 vaccines has been highlighted [63,64,65], with consideration of the higher levels of vaccine hesitancy found among females across several surveys [66].

To our knowledge, this is the first population study evaluating the effectiveness of SARS-CoV-2 vaccines with a two-year follow-up, providing relevant information on the duration of vaccine-induced protection, which will, however, need to be confirmed by additional studies on other populations. Other strengths of this study included the use of official, routinely collected electronic health databases from the entire general population, and the opportunity to adjust the estimates for multiple comorbidities, thus reducing the risk of misclassification related to defining COVID-19-related deaths [67].

This study has also limitations that must be considered when interpreting the results. First, as already discussed, testing policies and restrictive measures applied in Italy likely influenced the results on vaccine effectiveness against infection. Second, we did not have the individual laboratory data to specifically investigate vaccine effectiveness across different variants; however, all data related to the effectiveness of booster doses mostly refer to the Omicron variant and its descendent lineages, which have been dominating the landscape since January 2022 [68]. Third, a limitation is inherent in the study design, as negative test case–control designs are increasingly used to assess vaccine effectiveness, mainly due to their capacity to reduce bias from differential healthcare-seeking behavior between cases and controls [69]. However, it has been argued that in the context of COVID-19, the test-negative studies might be prone to bias caused by specific testing strategies or behaviors at the study location [10]. Fourth, the observational study design did not allow for a proper head-to-head comparison of diverse vaccines’ effectiveness [70]. Fifth, no updated information was available from the regional surveillance network about the serious adverse events in the studied population, and we were thus unable to evaluate the safety profile of the adopted vaccines in this follow-up. However, three studies previously published on the same population showed an acceptable safety profile of vaccination with up to 18 months of follow-up [12,25,32]. Finally, we based our definition of infection on the available laboratory information, which collected all positive swabs, but the infection rate is certainly underestimated, as the monitoring system could not detect all asymptomatic infections [71,72]. Although this issue might markedly change the estimates of the SARS-CoV-2 case-fatality rate [73], its impact on the comparison between unvaccinated and vaccinated subjects remains unknown, as it has not been determined whether the rate of undetected infections varies by vaccination status [74].

## 5. Conclusions

This large population-based cohort study confirms the effectiveness of COVID-19 vaccines against severe and lethal COVID-19 disease after two years of the immunization campaign, with remarkably higher levels of protection among the subjects who received one or more booster doses. A massive benefit of vaccination was documented for the elderly, but no protection against infection was observed, although the latter finding was unavoidably influenced by the Italian restriction policies to control the pandemic. Further studies on different populations are required to confirm these findings.

## Figures and Tables

**Figure 1 vaccines-11-01325-f001:**
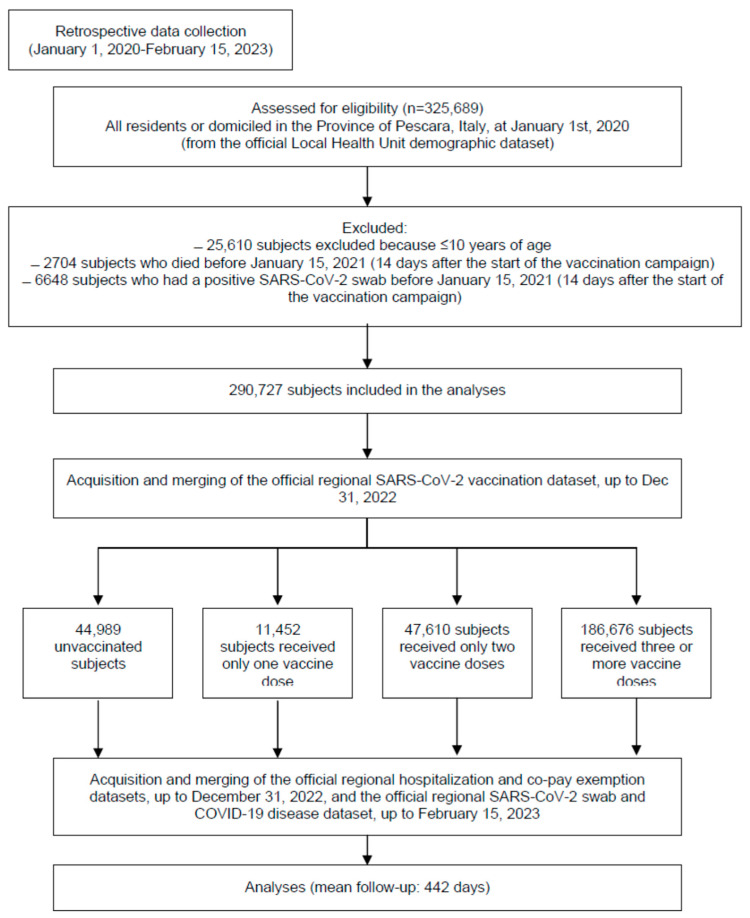
The flowchart of the participants according to the STROBE (Strengthening the Reporting of Observational Studies in Epidemiology) statement (http://www.strobe-statement.org, accessed on 3 August 2023).

**Table 1 vaccines-11-01325-t001:** Main characteristics of the sample, overall and by vaccine status.

	Unvaccinated	1 Dose ^A^	2 Doses ^B^	3/4 Doses ^C^	Total Sample
	(*n* = 44,989)	(*n* = 11,452)	(*n* = 47,610)	(*n* = 186,676)	(*n* = 290,727)
Mean age in years (SD)	44.4 (20.2)	41.9 (19.3)	40.7 (20.3)	52.5 (20.2)	48.9 (20.8)
Gender	%	%	%	%	% (*n*)
Females	14.6	3.8	16.2	65.5	51.2 (148,770)
Males	16.4	4.1	16.6	62.9	48.8 (141,957)
Age class in years					
10–29	18.5	5.8	26.2	49.5	21.7 (63,174)
30–59	17.3	4.2	16.8	61.7	47.1 (136,875)
60 or more	10.6	2.3	8.8	78.3	31.2 (90,678)
Risk factors and comorbidities ^D^					
No hypertension	16.7	4.2	17.5	61.6	86.2 (250,472)
Hypertension	8.1	2.1	9.7	80.1	13.9 (40,255)
No diabetes	15.9	4.0	16.7	63.4	94.6 (275,128)
Diabetes	8.8	2.5	10.0	78.7	5.4 (15,599)
No CVD	16.0	3.9	16.8	63.3	92.0 (267,475)
CVD	10.0	4.1	11.5	74.5	8.0 (23,252)
No COPD	15.6	3.9	16.4	64.0	96.2 (279,692)
COPD	11.2	3.7	15.5	69.6	3.8 (11,035)
No kidney disease	15.5	4.0	16.4	64.1	98.1 (285,296)
Kidney disease	13.1	2.8	14.3	69.8	1.9 (5431)
No cancer	15.9	4.0	16.7	63.4	94.3 (274,147)
Cancer	9.1	2.3	11.3	77.3	5.7 (16,580)
Type of vaccine ^E^					
BNT162b2	--	52.1	68.5	38.6	45.1 (110,720)
mRNA-1273	--	45.6	22.0	11.0	14.8 (36,259)
ChAdOx1 nCoV-19	--	2.0	3.7	0.0	0.8 (2007)
JNJ-78436735	--	--	1.1	0.0	0.2 (508)
NVX-CoV2373	--	0.0	4.4	0.0	0.1 (144)
Mixed ^F^	--	0.3	0.2	50.4	39.1 (96,100)
Mean follow-up in days (SD) ^G^	750 (123)	404 (107)	472 (108)	400 (52)	466 (148)

SD = Standard deviation. ^A^ Individuals who received only one dose of mRNA-1273, BNT162b2, ChAdOx1 nCoV-19, or NVX-CoV2373 vaccines between 2 January 2021 and 31 December 2022. ^B^ Individuals who received only one dose of JNJ-78436735 or two doses of the other vaccines between 2 January 2021 and 31 December 2022. ^C^ Subjects who received three or four doses of BNT162b2, mRNA-1273, ChAdOx1 nCoV-19, or NVX-CoV2373 vaccines or two or more doses of JNJ-78436735 between 2 January 2021 and 31 December 2022. ^D^ Details are available in the Methods section. ^E^ Column percentages have been reported for this variable. ^F^ Individuals who received ≥2 different vaccines. ^G^ The follow-up ended on the date of death or 15 February 2023. The start of follow-up varied across vaccine categories: (a) two weeks after the single dose of vaccine, if only one dose was administered; (b) two weeks after the second dose, if only two doses were administered; (c) 14 days after the third dose if three or more vaccine doses were administered; (d) 15 January 2021 for the group “unvaccinated”.

**Table 3 vaccines-11-01325-t003:** Adjusted hazard ratios (HR; 95% confidence interval—CI) ^A^ of the outcomes of vaccination effectiveness, overall and by age category.

Outcomes	SARS-CoV-2	COVID-19 ^B^	COVID-19-Related Death ^B^	All-Cause Death
	HR (95% CI)	HR (95% CI)	HR (95% CI)	HR (95% CI)
Vaccine doses				
Unvaccinated	1 (Ref. cat.)	1 (Ref. cat.)	1 (Ref. cat.)	1 (Ref. cat.)
1 dose ^C^	1.26 (1.21–1.32) *	0.47 (0.37–0.60) *	0.36 (0.28–0.47) *	1.40 (1.24–1.58) *
2 doses ^D^	2.41 (2.37–2.46) *	0.27 (0.24–0.30) *	0.38 (0.32–0.44) *	1.36 (1.28–1.45) *
3/4 doses ^E^	1.27 (1.25–1.29) *	0.12 (0.11–0.13) *	0.15 (0.14–0.17) *	0.22 (0.20–0.23) *
**Age class**, years				
60 or more				
Unvaccinated	1 (Ref. cat.)	1 (Ref. cat.)	1 (Ref. cat.)	1 (Ref. cat.)
1 dose ^C^	1.07 (0.95–1.20)	0.58 (0.44–0.77) *	0.36 (0.27–0.48) *	1.40 (1.23–1.59) *
2 doses ^D^	1.91 (1.82–2.00) *	0.35 (0.31–0.40) *	0.43 (0.37–0.50) *	1.45 (1.36–1.56) *
3/4 doses ^E^	1.09 (1.05–1.13) *	0.14 (0.13–0.16) *	0.16 (0.14–0.18) *	0.22 (0.20–0.23) *
30–59				
Unvaccinated	1 (Ref. cat.)	1 (Ref. cat.)	1 (Ref. cat.)	1 (Ref. cat.)
1 dose ^C^	1.46 (1.38–1.56) *	0.33 (0.20–0.53) *	0.29 (0.14–0.60) *	1.19 (0.82–1.72)
2 doses ^D^	2.73 (2.66–2.81) *	0.16 (0.13–0.19) *	0.12 (0.07–0.23) *	0.78 (0.61–1.00)
3/4 doses ^E^	1.46 (1.43–1.50) *	0.08 (0.07–0.10) *	0.11 (0.07–0.17) *	0.20 (0.16–0.25) *
10–29				
Unvaccinated	1 (Ref. cat.)	1 (Ref. cat.)	1 (Ref. cat.)	1 (Ref. cat.)
1 dose ^C^	1.13 (1.05–1.22) *	0.25 (0.03–1.86)	NE	1.57 (0.49–4.97)
2 doses ^D^	2.10 (2.03–2.17) *	0.21 (0.11–0.40) *	NE	0.17 (0.04–0.76) **
3/4 doses ^E^	1.01 (0.98–1.04)	0.17 (0.08–0.33) *	0.20 (0.03–1.24)	0.22 (0.08–0.59) **

NE = Not estimable. ^A^ Based on Cox proportional hazards models. The unvaccinated subjects are the reference group for all analyses. ^B^ Analyses restricted to the infected subjects. ^C^ Please see Table 1 footnote A. ^D^ Please see Table 1 footnote B. ^E^ Please see Table 1 footnote C. * *p* < 0.001; ** *p* < 0.05.

## Data Availability

The data presented in this study are available upon reasonable request from the corresponding author.

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
