# Peer review of "COVID-19 Vaccination Effectiveness in the General Population of an Italian Province: Two Years of Follow-Up"

_vaccines, 2023, doi:10.3390/vaccines11081325_

Round 1
Reviewer 1 Report
In this work authors present a retrospective cohort study evaluating COVID-19 vaccination effectiveness in general population of an Italian Province, up to two years from the beginning vaccination campaign in January 2021. The aim is to compare the risk of SARS‐Cov‐2 infection, severe or lethal COVID‐19 of the vaccinated versus unvaccinated subjects that received one, two or three/four doses.
The manuscript reports data in an excellent way, methods seems complete and clearly reported, findings are perfectly summarized in tables, the discussion is complete and cites appropriate recent literature.
Minor editing of English language required.
Author Response
We thank the Reviewer for his/her comments.
Reviewer 2 Report
There are different immune deficiency conditions - what case definition was used in your research? Recommend to include: vaccine effectiveness a) in healthy population control group, b) during omicron predominance, c) against hospital admission.
Author Response
II-1. The Reviewer wrote: "There are different immune deficiency conditions - what case definition was used in your research?"
We absolutely agree that accounting for the pre-existing immunological status would have added useful insights to the present findings. Unfortunately, however, we had no data on the immune deficiency conditions of the included subjects. Given to the initial protocol restrictions, we were able to extract only the conditions that have been listed in the manuscript (diabetes, hypertension, major cardiovascular or cerebrovascular diseases, chronic obstructive pulmonary diseases, kidney diseases, and all cancer).
II-2. The Reviewer wrote: "Recommend to include: vaccine effectiveness a) in healthy population control group, b) during Omicron predominance, c) against hospital admission".
We entirely agree that reporting separate results during the Omicron predominance would significantly improve the analysis, and we accordingly added a separate supplementary analysis (reported in Table S3) restricted to the period of the Omicron predominance (from 1 January 2022 to the end of follow-up). In these analyses, the subjects who died or were infected with SARS-CoV-2 before 1 January 2022 were excluded, and the follow-up started on 1 January, 2022 for all the subjects who were not vaccinated or had their last dose administered before that date. Please acknowledge that the Methods, Results and Discussion were accordingly revised to explain the new analyses and discuss their results.
We also entirely agree on the utility to add some data on a healthy population control group, and accordingly added a new set of analyses restricted to the subjects who had none of the recorded comorbidities and were not hospitalized for any cause during the three years preceding the pandemic (2018-2020). Please acknowledge that all the results – shown in the new supplemental Table S1 – were similar to those of the overall sample. We reported the new analyses and their results in the Methods and Results.
As regards the effectiveness of vaccines against hospital admission, we entirely agree that this is a pivotal issue. Please acknowledge that we included the hospital admissions in the definition of the “severe COVID-19 outcome”, that was defined as “a severe disease, diagnosed by a specialist physician, virologically confirmed, and requiring hospital admission".
Reviewer 3 Report
This is a retrospective population study on the effectiveness of COVID-19 vaccination in the Italian province of Pescara. The advantages of the study are the large size (nearly 300 000 people) and long follow-up (over 2 years) added with accurate Italian statistics.
At the same time, some factors may confound the results and confuse the interpretation, and, hence, reduce the significance of findings and conclusions. Firstly, the pandemic period can be roughly divided into two: January to December 2021 with dominance of the original coronavirus and delta (and other) variants, and from December 2021 to the end of study period in February 2023 with dominance of omicron variants. Omicron by definition was more contagious but less fatal, and this might be reflected in vaccine effectiveness against infection and death, respectively. Somehow this time factor should be considered in the analysis and discussion. Many different vaccines seem to have been used in Italy. In this analysis, all vaccines are pooled assuming that they are equal. This may be largely true, but an attempt should be made to prove it in the present large material.
One of the most interesting and potentially important findings of the present study is related to diverse vaccines. The results suggest that a mixed regimen is more effective than the same number of doses of one vaccine against COVID-19 disease (Table 4) and death (Table 5). This should be analysed in more detail to confirm or refute.
Death is of course the most important end point, and reduction in overall mortality in this study is quite impressive. However, the reduction should be presented, as it is in the Results, also in the Abstract and Discussion by saying that mortality in ≥60 years was 22% in unvaccinated and 3% in the vaccinated persons, not by saying that reduction was 19%, which does not mean anything.
The authors have collected information about underlying diseases, but use this information only in the comparison of groups (Table 1). Because the numbers are quite large for hypertension, diabetes and cancer, it should be possible to analyse vaccine effectiveness also in these critical subgroups. That might add value to the study.
Minor editing of English language required
Author Response
III-1. The Reviewer wrote: "This is a retrospective population study on the effectiveness of COVID-19 vaccination in the Italian province of Pescara. The advantages of the study are the large size (nearly 300 000 people) and long follow-up (over 2 years) added with accurate Italian statistics. At the same time, some factors may confound the results and confuse the interpretation, and, hence, reduce the significance of findings and conclusions. Firstly, the pandemic period can be roughly divided into two: January to December 2021 with dominance of the original coronavirus and delta (and other) variants, and from December 2021 to the end of study period in February 2023 with dominance of omicron variants. Omicron by definition was more contagious but less fatal, and this might be reflected in vaccine effectiveness against infection and death, respectively. Somehow this time factor should be considered in the analysis and discussion".
We entirely agree that reporting separate results during the Omicron predominance would significantly improve the analysis, and we accordingly added a separate supplementary analysis (reported in Table S3) restricted to the period of the Omicron predominance (from 1 January 2022 to the end of follow-up). In these analyses, the subjects who died or were infected with SARS-CoV-2 before 1 January 2022 were excluded, and the follow-up started on 1 January, 2022 for all the subjects who were not vaccinated or had their last dose administered before that date. Please acknowledge that the Methods, Results and Discussion were accordingly revised to explain the new analyses and discuss their results.
Concerning the Abstract, we added the following sentence: “Importantly, during the Omicron predominance period only the group who received at least a booster dose showed a reduced risk of COVID-19 related death.”
As regards the Results, the following paragraph was added: “When the analyses were repeated restricting the follow-up to the Omicron predominance period (Table S3), both univariate and multivariable results were concordant to those of the overall sample, with the notable exception of a lower protection among the subjects who received a single or two vaccine doses only. Against COVID-19, the adjusted HRs of the recipients of one or two doses were 0.76 and 0.63, respectively (versus 0.47 and 0.27 during the entire follow-up). Most importantly, no protection was observed for these subjects against COVID-19 related deaths (both p>0.05). Thus, during the Omicron predominance period, only the individuals who received at least one booster dose showed a significantly and substantially lower risk of lethal COVID-19.”
Concerning the Discussion, we added the following sentences: “The impact of booster doses on the protection against severe disease or death was even larger when the analyses were restricted to the Omicron predominance period, when the effectiveness of the single or two doses schedules were substantially lower than during the overall follow-up, and only the subjects who received at least one booster dose showed a significant decrease in the risk of a lethal COVID-19.”
III-2. The Reviewer wrote: "Many different vaccines seem to have been used in Italy. In this analysis, all vaccines are pooled assuming that they are equal. This may be largely true, but an attempt should be made to prove it in the present large material".
The reviewer also wrote: "One of the most interesting and potentially important findings of the present study is related to diverse vaccines. The results suggest that a mixed regimen is more effective than the same number of doses of one vaccine against COVID-19 disease (Table 4) and death (Table 5). This should be analysed in more detail to confirm or refute".
We agree and accordingly performed multivariable analyses, stratified by vaccine doses, to compare the effectiveness of the three most frequently administered vaccine schedule: BNT162b2, mRNA-1273 and mixed. Unvaccinated subjects were excluded from these analyses, which compared directly the likelihood of each outcome among those who received only mRNA-1273 vaccine, or mixed vaccines, versus those who received only BNT162b2 (as the reference category). The results were reported in the new Table S4. Please acknowledge that the Methods, Results, and Discussion have been accordingly revised to explain the new analyses and discuss their results.
III-3. The Reviewer wrote: "Death is of course the most important end point, and reduction in overall mortality in this study is quite impressive. However, the reduction should be presented, as it is in the Results, also in the Abstract and Discussion by saying that mortality in ≥60 years was 22% in unvaccinated and 3% in the vaccinated persons, not by saying that reduction was 19%, which does not mean anything".
We entirely agree, and thank the Reviewer. Accordingly, please acknowledge that the sentence in the Abstract "A massive impact of vaccination was found among the elderly, with a 19% absolute risk reduction of COVID-19-related death among the recipients of booster doses" was replaced with "A massive impact of vaccination was found among the elderly: 22.0% of the unvaccinated, infected individuals died, as opposed to less than 3% of those who received ≥3 vaccine doses".
III-4. The Reviewer wrote: "The authors have collected information about underlying diseases, but use this information only in the comparison of groups (Table 1). Because the numbers are quite large for hypertension, diabetes and cancer, it should be possible to analyse vaccine effectiveness also in these critical subgroups. That might add value to the study".
We agree and accordingly added the multivariate analyses restricted to the subgroups of subjects with hypertension, diabetes or cancer, for each outcome, in the new Table S2. We mentioned the new analyses and their results in the Methods and the Results: overall, the results for all subgroups were similar to those of the overall sample.
Reviewer 4 Report
1.Please provide more research background.
2.The annotations in Table 1 show: “ When not otherwise stated, the values in the tables are reported as % (n)”,but except for total sample, all other values only show% without n.
3.The annotations in Table 1 show: “A Individuals who received only one dose of mRNA‐1273, BNT162b2, ChAdOx1 nCoV‐19, JNJ‐78436735 or NVX‐CoV2373 vaccines between January 2, 2021 and December 31, 2022. B Individuals who received only one dose of JNJ‐78436735, or two doses of the other vaccines, between January 2, 2021 and December 31, 2022.. C Subjects who received three or four doses of BNT162b2, mRNA‐1273, ChAdOx1 nCoV‐19, JNJ‐78436735 or NVX‐CoV2373 vaccines between January 2, 2021 and December 31, 2022. D Details are available in the Methods section. ”.
But in fact, no subject in the 1 Dose group and 3/4 Doses group received JNJ-78436735 vaccines. Why are those individuals who received only one dose of JNJ‐78436735 divided into 2 Dose group? But they were also assigned to the 1 Dose group according to Note A.
4.Table 2, Table 4, Table 5, Table 6: Please annotate the statistical analysis results to show whether there are statistical differences between the results of each group.
5.Table 2. Overall sample Positive swabs, % (n); Table 4. Overall sample COVID‐19, % (n); Table 5 Overall sample Deaths, % (n): Why is there only% without (n) in the 1 dose , 2 dose and 3/4 Doses groups?
Minor editing of English language required.
Author Response
IV-1. The Reviewer wrote: "Please provide more research background".
We agree with the Reviewer with the need to provide more research background. We accordingly included in the Introduction the following additional, important references to recent population studies assessing the effectiveness of booster doses in restoring COVID-19 vaccine protection:
- Accorsi, E.K.; Britton, A.; Fleming-Dutra, K.E.; Smith, Z.R.; Shang, N.; Derado, G.; Miller, J.; Schrag, S.J.; Verani, J.R. Association Between 3 Doses of mRNA COVID-19 Vaccine and Symptomatic Infection Caused by the SARS-CoV-2 Omicron and Delta Variants. Jama 2022, 327, 639-651, doi:10.1001/jama.2022.0470.
- Atanasov, V.; Barreto, N.; Whittle, J.; Meurer, J.; Weston, B.W.; Luo, Q.E.; Franchi, L.; Yuan, A.Y.; Zhang, R.; Black, B. Understanding COVID-19 Vaccine Effectiveness against Death Using a Novel Measure: COVID Excess Mortality Percentage. Vaccines (Basel) 2023, 11, doi:10.3390/vaccines11020379.
- Tartof, S.Y.; Slezak, J.M.; Puzniak, L.; Hong, V.; Frankland, T.B.; Ackerson, B.K.; Takhar, H.S.; Ogun, O.A.; Simmons, S.R.; Zamparo, J.M.; et al. Effectiveness of a third dose of BNT162b2 mRNA COVID-19 vaccine in a large US health system: A retrospective cohort study. Lancet Reg Health Am 2022, 9, 100198, doi:10.1016/j.lana.2022.100198.
- Thompson, M.G.; Natarajan, K.; Irving, S.A.; Rowley, E.A.; Griggs, E.P.; Gaglani, M.; Klein, N.P.; Grannis, S.J.; DeSilva, M.B.; Stenehjem, E.; et al. Effectiveness of a Third Dose of mRNA Vaccines Against COVID-19-Associated Emergency Department and Urgent Care Encounters and Hospitalizations Among Adults During Periods of Delta and Omicron Variant Predominance - VISION Network, 10 States, August 2021-January 2022. MMWR Morb Mortal Wkly Rep 2022, 71, 139-145, doi:10.15585/mmwr.mm7104e3.
- Mateo-Urdiales, A.; Sacco, C.; Fotakis, E.A.; Del Manso, M.; Bella, A.; Riccardo, F.; Bressi, M.; Rota, M.C.; Petrone, D.; Siddu, A.; et al. Relative effectiveness of monovalent and bivalent mRNA boosters in preventing severe COVID-19 due to omicron BA.5 infection up to 4 months post-administration in people aged 60 years or older in Italy: a retrospective matched cohort study. Lancet Infect Dis 2023, doi:10.1016/S1473-3099(23)00374-2.
- Andrews, N.; Stowe, J.; Kirsebom, F.; Toffa, S.; Sachdeva, R.; Gower, C.; Ramsay, M.; Lopez Bernal, J. Effectiveness of COVID-19 booster vaccines against COVID-19-related symptoms, hospitalization and death in England. Nat Med 2022, 28, 831-837, doi:10.1038/s41591-022-01699-1.
IV-2. The Reviewer wrote: "The annotations in Table 1 show: “When not otherwise stated, the values in the tables are reported as % (n)", but except for total sample, all other values only show % without n".
The Reviewer also wrote: "The annotations in Table 1 show: “A Individuals who received only one dose of mRNA-1273, BNT162b2, ChAdOx1 nCoV-19, JNJ-78436735 or NVX-CoV2373 vaccines between January 2, 2021 and December 31, 2022. B Individuals who received only one dose of JNJ-78436735, or two doses of the other vaccines, between January 2, 2021 and December 31, 2022. C Subjects who received three or four doses of BNT162b2, mRNA-1273, ChAdOx1 nCoV-19, JNJ-78436735 or NVX-CoV2373 vaccines between January 2, 2021 and December 31, 2022. D Details are available in the Methods section.”. But in fact, no subject in the 1 Dose group and 3/4 Doses group received JNJ-78436735 vaccines. Why are those individuals who received only one dose of JNJ-78436735 divided into 2 Dose group? But they were also assigned to the 1 Dose group according to Note A ".
We entirely agree that Table 1 footnote was not precise and we reported the symbols “%” or “% (n)” in the correct columns.
With regard to the footnote "A" (referring to the individuals who received only one vaccine dose), we incorrectly specified it; please accept our apologies for the error and for the confusion. Please acknowledge that, in the first year of the pandemic, a single dose of JNJ-78436735 vaccine was considered equivalent to two doses of all the other vaccines, thus the subjects who received a single dose of JNJ-78436735 were included into the "two doses" group. Thus, we removed JNJ-78436735 vaccine from the point "A" of the footnote. This was a writing oversight, thank you very much for the correction.
With regard to the "3/4 doses" group, it was possible to receive a second dose of JNJ-78436735 vaccine (which is equivalent to a 3rd dose of the other vaccine types), although none in our sample did (very few doses were available after the first phases of the pandemic). Thus, in this case we listed JNJ-78436735 into the footnote "C" among the vaccines administered to the group "≥3 doses", but we incorrectly specified the peculiar number of doses (2 or more) of the subjects immunized with this type of vaccine. As such, please acknowledge that the footnote "C" was amended, from "Subjects who received three or four doses of BNT162b2, mRNA-1273, ChAdOx1 nCoV-19, JNJ-78436735 or NVX-CoV2373 vaccines" into "Subjects who received three or four doses of BNT162b2, mRNA-1273, ChAdOx1 nCoV-19, or NVX-CoV2373 vaccines, or two or more doses of JNJ-78436735".
We thank the Reviewer and, again, apologize for the oversights and the confusion.
IV-3. The Reviewer wrote: "Table 2, Table 4, Table 5, Table 6: Please annotate the statistical analysis results to show whether there are statistical differences between the results of each group".
We agree and accordingly added, in each Table, the p-values that were >0.05 (all others were significant, thus we tried to reduce redundancy).
IV-4. The Reviewer wrote: "Table 2. Overall sample Positive swabs, % (n); Table 4. Overall sample COVID‐19, % (n); Table 5. Overall sample Deaths, % (n): Why is there only % without (n) in the 1 dose, 2 dose and 3/4 Doses groups?"
We entirely agree and accordingly added the corresponding raw data in all tables. Again, we thank You very much for the correction.